

# Species–specific crab predation on the hydrozoan clinging jellyfish *Gonionemus* sp. (Cnidaria, Hydrozoa), subsequent crab mortality, and possible ecological consequences

Mary R. Carman[1], David W. Grunden[2] and Annette F. Govindarajan[1]

[1] Biology Department, Woods Hole Oceanographic Institution, Woods Hole, MA, United States of America
[2] Town of Oak Bluffs Shellfish Department, Oak Bluffs, MA, United States of America

## ABSTRACT

Here we report a unique trophic interaction between the cryptogenic and sometimes highly toxic hydrozoan clinging jellyfish *Gonionemus* sp. and the spider crab *Libinia dubia*. We assessed species–specific predation on the *Gonionemus* medusae by crabs found in eelgrass meadows in Massachusetts, USA. The native spider crab species *L. dubia* consumed *Gonionemus* medusae, often enthusiastically, but the invasive green crab *Carcinus maenus* avoided consumption in all trials. One out of two blue crabs (*Callinectes sapidus*) also consumed *Gonionemus*, but this species was too rare in our study system to evaluate further. *Libinia* crabs could consume up to 30 jellyfish, which was the maximum jellyfish density treatment in our experiments, over a 24-hour period. *Gonionemus* consumption was associated with *Libinia* mortality. Spider crab mortality increased with *Gonionemus* consumption, and 100% of spider crabs tested died within 24 h of consuming jellyfish in our maximum jellyfish density containers. As the numbers of *Gonionemus* medusae used in our experiments likely underestimate the number of medusae that could be encountered by spider crabs over a 24-hour period in the field, we expect that *Gonionemus* may be having a negative effect on natural *Libinia* populations. Furthermore, given that *Libinia* overlaps in habitat and resource use with *Carcinus*, which avoids *Gonionemus* consumption, *Carcinus* populations could be indirectly benefiting from this unusual crab–jellyfish trophic relationship.

## INTRODUCTION

Gelatinous zooplankton are important and often conspicuous members of many marine communities, but blooms are often problematic as they may interfere with fisheries and aquaculture, clog power plant intake pipes, and present sting risks to humans (*Purcell, Uye & Lo , 2007*; *Graham & Bayha, 2008*). Anthropogenic activities have contributed to the spread of jellyfish outside their native range (*Purcell, Uye & Lo , 2007*; *Graham & Bayha, 2008*), where they can also have negative consequences to the ecosystem

Corresponding author
Annette F. Govindarajan,
afrese@whoi.edu

(*Manzari et al., 2015*). A likely potential impact of invasive jellyfish is through alteration of native food webs, often thought to manifest through predation and competition (*Pauly et al., 2009*; *Graham & Bayha, 2008*). Jellyfish are less often thought of as prey (*Arai & Jacobs, 1980*; *Arai, 2005*; *Ates, 2017*) and are sometimes assumed to be trophic dead-ends (*Sommer et al., 2002*; *Lynam et al., 2006*; *Yamamoto et al., 2008*; *Condon et al., 2011*), but this paradigm is changing (*Cardona et al., 2012*; *Diaz-Briz et al., 2017*; *McInnes et al., 2017*).

"Gelata" is a general term that refers to phylogenetically diverse gelatinous zooplankton, including members of the phylum Cnidaria belonging to the Scyphozoa, Cubozoa, and Hydrozoa (collectively known as the Medusozoa), the phylum Ctenophora (ctenophores), and the phylum Chordata (salps, doliolids, and pyrosomes) (*Haddock, 2004*). Of these groups, most research has focused on a relatively small number of conspicuous scyphozoans (*Purcell, Uye & Lo , 2007*). Despite the relative lack of attention, the Hydrozoa is by far the most speciose and diverse group with around 842 valid medusa (i.e., jellyfish)—producing species (*Bouillon & Boero, 2000a*). The Hydrozoa is phylogenetically well-supported (*Collins et al., 2006*; *Kayal et al., 2013*; *Zapata et al., 2015*) and is sometimes referred to as a superclass (*Bouillon & Boero, 2000b*; *Xu et al., 2014*).

The clinging jellyfish *Gonionemus* sp. (Cnidaria, Hydrozoa, Limnomedusae; Fig. 1) is an increasingly conspicuous member of Northwest Atlantic eelgrass communities, and populations may be comprised of native and invasive lineages (*Govindarajan et al., 2017*). Like many cryptogenic species, insufficient taxonomy complicates our understanding of its biogeography (*Govindarajan et al., 2017*). Clinging jellyfish described as *Gonionemus murbachii* Mayer, 1901 (but later synonymized with *Gonionemus vertens* Agassiz, 1862) were first noted in Massachusetts and Connecticut in 1894, but nearly disappeared in the 1930s when its eelgrass habitat was decimated by a wasting disease (*Govindarajan & Carman, 2016*). In recent years, clinging jellyfish have made a comeback in these areas (*Govindarajan & Carman, 2016*).

*Gonionemus* lineages vary in their toxicity (*Naumov, 1960*), and some Sea of Japan populations are associated with stings that can cause severe pain, respiratory difficulty, paralysis, and other neurological symptoms, while populations in other parts of the world are harmless to humans (*Naumov, 1960*; *Otsuru et al., 1974*; *Yakovlev & Vaskovsky, 1993*). Nineteenth and early 20th century Northwest Atlantic *G. murbachii* populations were not associated with stings. However, painful stings similar to those associated with Sea of Japan populations began occurring in Massachusetts, USA, in 1990, suggesting an invasion of a new and highly toxic lineage (*Govindarajan & Carman, 2016*). Since then, clinging jellyfish blooms have been occurring regularly in Massachusetts, and the jellyfish appear to be expanding their range both inside and outside of Massachusetts (*Govindarajan & Carman, 2016*; *Gaynor et al., 2016*; *Govindarajan et al., 2017*).

*Govindarajan et al. (2017)* suggested that based on mitochondrial COI sequences and subtle morphological features that the Northwest Atlantic and Pacific forms (including highly toxic populations) were similar to each other, and different from *G. vertens* from the Northeast Pacific. It seems likely that the Northwest Atlantic/Northwest Pacific form is *G. murbachii* Mayer, 1901. However, a definitive link between past and contemporary

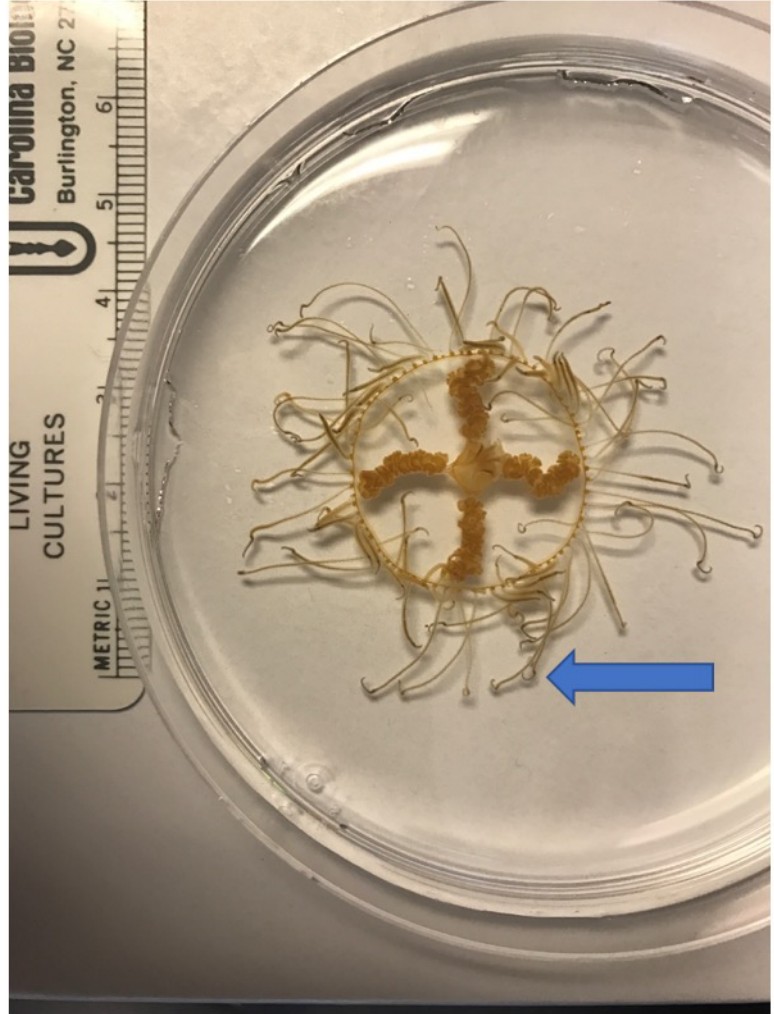

**Figure 1** **The clinging jellyfish *Gonionemus* sp.** The blue arrow points to the end of the tentacles where the adhesive structures are found.

populations is still lacking (*Govindarajan & Carman, 2016*; *Govindarajan et al., 2017*). Thus, we refer here to this form, which is our focal taxon in this study, as *Gonionemus* sp. (or simply "*Gonionemus*").

Clinging jellyfish are found primarily in eelgrass meadows, where they "cling" to eelgrass blades using the adhesive structures on their tentacles (*Naumov, 1960*; Fig. 1). Adult medusae typically range in size from 1–2.5 cm (*Govindarajan et al., 2017*) and feed on a variety of small zooplankton such as amphipods and isopods (*Yakovlev & Vaskovsky, 1993*). They are not known to have any predators, although molluscs may feed on the minute polyp life cycle stage (*Yakovlev & Vaskovsky, 1993*). The highly toxic nature of some *Gonionemus* lineages might act as a deterrent to potential predators, but it is also possible that predation on clinging jellyfish has been overlooked.

Northwest Atlantic eelgrass meadows are also home to predatory native and invasive crab species (*Able et al., 2002*; *Garbary et al., 2014*; *Neckles, 2015*; *Matheson et al., 2016*). We investigated the possibility that crabs can prey on *Gonionemus*, and the potential impact of *Gonionemus* prey on crab predators. The Massachusetts, USA eelgrass beds where *Gonionemus* medusae are found are home to native spider crabs (*Libinia dubia* Milne Edwards, 1834) and, occasionally, blue crabs (*Callinectes sapidus* Rathbun, 1896), and the invasive green crab (*Carcinus maenus* Linnaeus, 1758). Green crabs in particular are highly destructive to eelgrass ecosystems as they uproot eelgrass shoots while foraging and may graze directly on the eelgrass shoots (*Malyshev & Quijón, 2011*; *Garbary et al., 2014*). All three crab species feed on a wide variety of invertebrates (*Aldrich, 1974*; *Grosholz & Ruiz, 1996*; *Harding, 2003*; *Baeta et al., 2006*). While predation on jellyfish is often not considered (*Arai, 2005*), *Carcinus maenus* (*Lauckner, 1980*), *Callinectes sapidus* (*Farr, 1978*), and *Libinia dubia* (*Phillips, Burke & Keener, 1969*) have been reported to feed on scyphozoan medusae.

Our results demonstrated a new trophic interaction between crabs and a highly toxic hydrozoan jellyfish with consequences for invasive species impacts in ecologically sensitive eelgrass meadows. We found that the native spider and blue crabs consumed *Gonionemus*, but that the invasive green crabs did not. We further found that *Gonionemus* ingestion resulted in crab death when large numbers of jellyfish were consumed; however, blue crabs were too rare at our site to be assessed at higher jellyfish densities. Thus, we hypothesize that *Gonionemus* may potentially impact native ecosystems via differential predation by a native species (spider crabs) that may lead to a decline of that species, while avoidance of *Gonionemus* by a notoriously destructive invasive species (green crabs) may facilitate its dominance.

## MATERIAL & METHODS

### Study area

The experimental animals in our study were obtained from Farm Pond (41.44756, −70.55694) and Lagoon Pond (41.44816, −70.59022), which are semi-enclosed coastal ponds that harbor eelgrass beds on the northeastern side of the island of Martha's Vineyard in Massachusetts, USA (Fig. 2). Lagoon Pond covers 544 acres with a mean depth of 3 m, and Farm Pond covers 33 acres, is tidally restricted, and has a mean depth of 1.5 m. Both ponds have a tidal range of <1 m. The ponds are located in the town of Oak Bluffs, separated by about 4 km of land, and are the sites of ongoing research on invasive species (*Carman, Grunden & Ewart, 2014*; *Carman et al., 2016*; *Colarusso et al., 2016*). *Gonionemus* was first observed in Farm Pond in 2007 (*Govindarajan & Carman, 2016*) and has not been observed in Lagoon Pond. Permission to collect animals at our field sight was obtained through D. Grunden (Oak Bluffs, Massachusetts Shellfish Constable; in accordance with Massachusetts General Laws Chapter 130 Section 98).

### Identification of predatory crab species

We conducted four trials during June and July 2016 to identify which, if any, local crab species prey on *Gonionemus*. Crabs were trapped in Farm and Lagoon Ponds the week
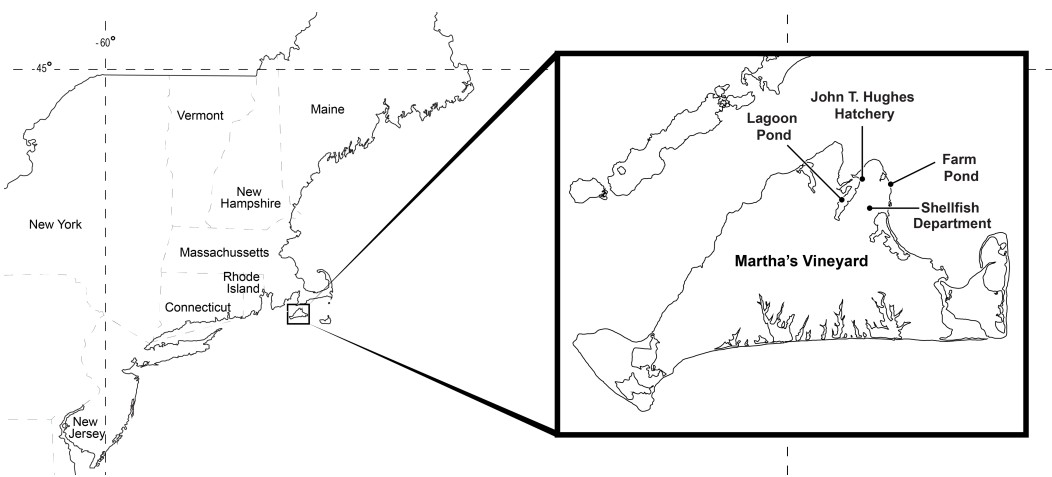

**Figure 2 Study locations.** Animals were collected at Lagoon Pond and Farm Pond, and experiments were conducted at the Oak Bluffs Shellfish Department and John T. Hughes Hatchery.

prior to each experiment using crab traps. The crabs were then kept in the cages for one week in a relatively barren area of Farm Pond that lacks *Gonionemus* habitat.

At the start of each experiment, individual crabs were transported in tubs of seawater to the laboratory at the Town of Oak Bluffs Shellfish Department in Massachusetts (Fig. 2). Crab size (carapace width) was recorded. At the same time as the crabs were removed from the crab traps, *Gonionemus* specimens were also collected from the eelgrass meadow in Farm Pond using hand held nets while wading and snorkeling, and transported along with the crabs to the laboratory.

Experiments were conducted in closed tubs (42 cm × 33 cm × 17 cm) of seawater. Five adult jellyfish (15–20 mm bell width) were placed in a tub with a single crab. Between two and six replicate tubs per crab species were set up on each sampling date, depending on the number of crabs that were caught (Table 1). Additionally, control tubs consisting of crabs only (with no jellyfish) and jellyfish only (with no crabs) were also set up for each experiment (Table 1). The number of jellyfish remaining and crab condition (dead or alive) were recorded at three time points (5 min, 3 h, and 24 h). We verified our assumption that jellyfish disappearances were due to predation by the crabs by: (1) direct observation of crabs consuming jellyfish, which we recorded by taking representative photographs and video; and (2) running jellyfish-only controls with each trial to assess jellyfish mortality independent of the crabs.

## Impact of jellyfish consumption on spider crabs

As a follow-up to our first set of trials which documented predation on jellyfish by *Libinia* (as well as the relatively rare *Callinectes sapidus*) and a possible association between jellyfish consumption and mortality, we assessed *Libinia* predation at higher jellyfish densities. We ran similar predation experiments on two dates in July 2017 at four additional jellyfish densities: 10, 15, 20, and 30 jellyfish per crab. The experiments were carried out in the

**Table 1  Experimental design and timeline of predation trials.**

| Trial date | Treatment | Crabs tested and # replicates |
|---|---|---|
| June 30, 2016 | 5 jellyfish | Green crabs—5<br>Spider crabs—5<br>Blue crabs—1 |
|  | 0 jellyfish | Green crabs—5<br>Spider crabs—5 |
|  | 5 jellyfish | No crabs—2 |
| June 30, 2016 | 5 jellyfish | Green crabs—5<br>Spider crabs—5<br>Blue crabs—1 |
|  | 0 jellyfish | Green crabs—2<br>Spider crabs—2 |
|  | 5 jellyfish | No crabs—2 |
| July 21, 2016 | 5 jellyfish | Green crabs—6<br>Spider crabs—6 |
|  | 0 jellyfish | Green crabs—6<br>Spider crabs—6 |
|  | 5 jellyfish | No crabs—2 |
| July 28, 2016 | 5 jellyfish | Green crabs—6<br>Spider crabs—6 |
|  | 0 jellyfish | Green crabs—2<br>Spider crabs—2 |
|  | 5 jellyfish | No crabs—2 |
| July 7, 2017 | 10 jellyfish | Spider crabs—6 |
|  | 0 jellyfish | No crabs—2 |
|  | 15 jellyfish | Spider crabs—6 |
|  | 0 jellyfish | No crabs—2 |
| July 18, 2017 | 20 jellyfish | Spider crabs—6 |
|  | 0 jellyfish | No crabs—2 |
|  | 30 jellyfish | Spider crabs—6 |
|  | 0 jellyfish | No crabs—2 |

laboratory at the Martha's Vineyard Shellfish Group, Inc.'s John T. Hughes Hatchery and Research Facility (leased from the Massachusetts Division of Marine Fisheries) in Oak Bluffs (Fig. 2). As with the 2016 experiments, crabs were trapped during the week before the experiment and held in Farm Pond without supplemental food. Also as in the 2016 trials, jellyfish were obtained from Farm Pond immediately prior to the start of the experiments. Crabs were placed in tubs with a given number of jellyfish (10, 15, 20, or 30 adult jellyfish); with 6 replicates per jellyfish density. Control tubs with crabs only and jellyfish only were also set up on each experiment date. The number of jellyfish remaining and crab condition (dead or alive) after 15 min and 24 h were recorded.

To confirm that the *Gonionemus* densities we used in our predation trials were realistic compared to what the crabs encounter in nature, jellyfish densities were recorded on three dates in 2017 by counting the number of jellyfish in representative 3 m × 3 m areas in the

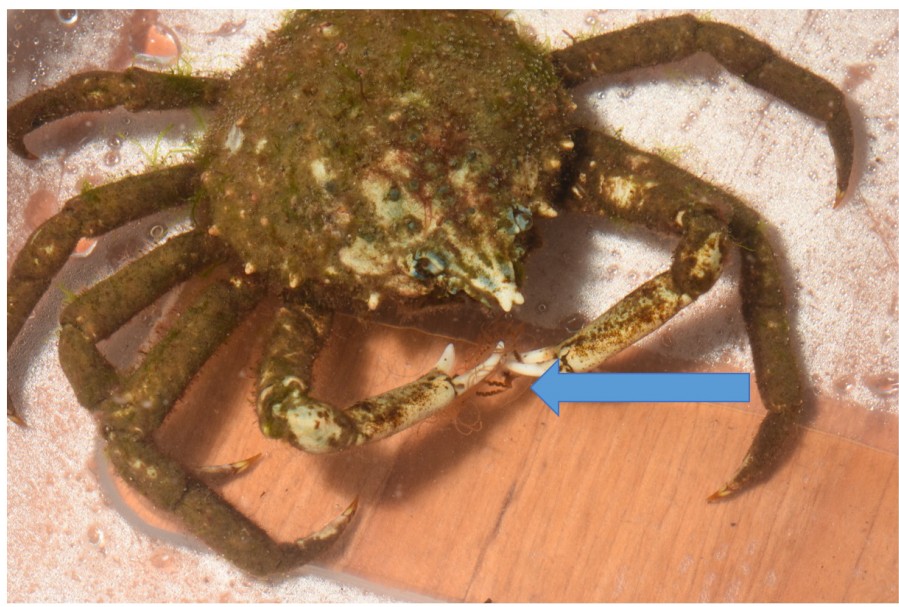

**Figure 3** **Predation on *Gonionemus*.** Spider crab using its claws to capture and consume a *Gonionemus* medusa (indicated by the blue arrow). Photograph by Dann Blackwood.

part of Farm Pond where the jellyfish are found. The jellyfish were collected by net scoops and counted. This method likely underestimates the true *Gonionemus* abundance, and so is a conservative depiction.

## RESULTS

### Identification of predatory crab species

Several spider crabs (*Libinia dubia*) and green crabs (*Carcinus maenas*) were collected in our crab traps, as well as 2 blue crabs (*Callinectes sapidus*). Mean carapace width was 62 mm $\pm$ 9 S.D. in *Libinia* ($n = 30$), 62 mm $\pm$ 6 S.D. in *Carcinus* ($n = 30$), and 63 mm and 78 mm in the two *Callinectes* individuals.

Twenty-one out of 22 spider crabs and one out of the 2 blue crabs obtained consumed *Gonionemus* (Figs. 3 and 4A), but none of the green crabs did. We observed *Libinia* predation on the jellyfish almost immediately at the start of our trials (Fig. 3). Often, spider crabs consumed 100% of the jellyfish, and most jellyfish consumption occurred within the first 3 h (Fig. 4A).

At the end of the 24-hour periods, *Libinia* mortality (27.3%) was higher than in the corresponding no—jellyfish controls (12.5%), and *Carcinus* trials with (5%) and without (12.5%) jellyfish. To assess the role of crab size on mortality, the 22 *Libinia* that received the jellyfish were sorted into three size (carapace width) categories: 50–58 mm, 60–69 mm, and 70–82 mm. The percent mortality increased with size category (Fig. 5). Each size category contained individuals used on all 4 of the trial dates (Data S1). For all trials, 100% of the jellyfish in the jellyfish-only control tubs were alive at the end of the 24-hour periods.
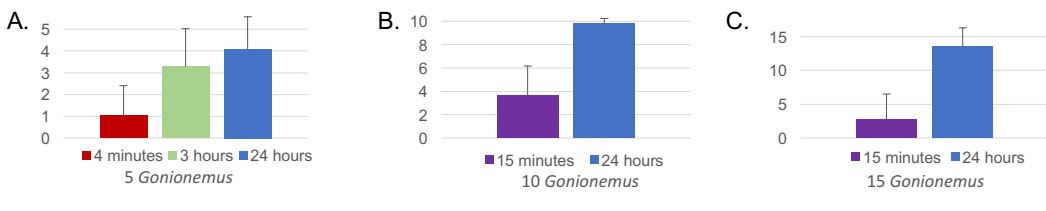

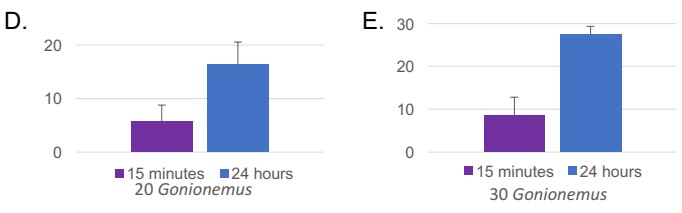

**Figure 4   Mean number of *Gonionemus* consumed at different *Gonionemus* densities and exposure times.** (A) *Gonionemus* density = 5/tub; (B) *Gonionemus* density = 10/tub; (C) *Gonionemus* density = 15/tub; (D) *Gonionemus* density = 20/tub; (E) *Gonionemus* density = 30/tub. Predation values are cumulative over the course of exposure. Error bars represent standard deviations. Note the differences in the $y$-axis scales for each graph. In each graph, the top gridline indicates the number of *Gonionemus* placed in each crab tub.

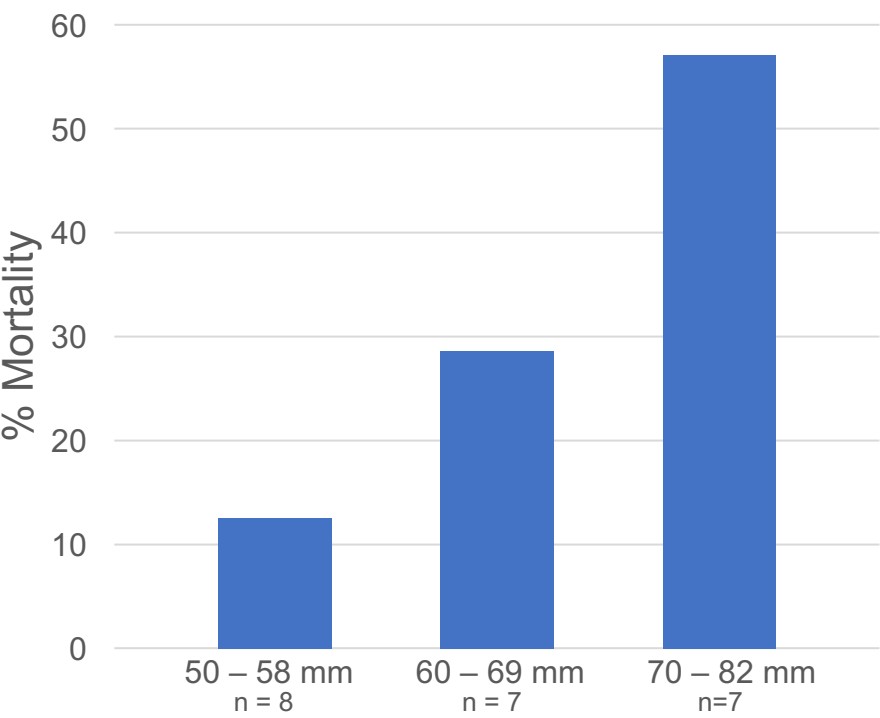

**Figure 5   *Libinia* mortality in each size class.** Data are from the *Libinia* used in the 5—*Gonionemus* trials.

A.

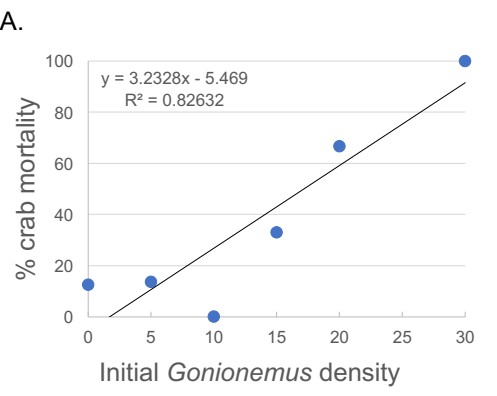

B.

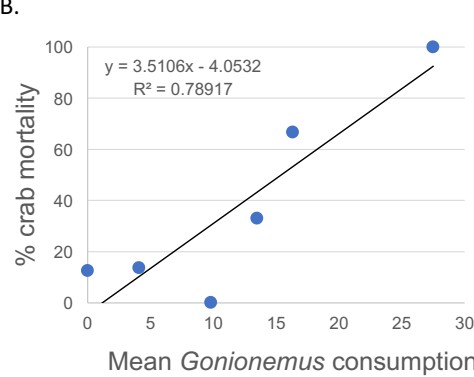

**Figure 6 Spider crab mortality and different *Gonionemus* densities and consumption levels.** (A) Mean *Libinia* mortality as a function of *Gonionemus* density (number of medusae initially placed in crab containers); and (B) *Gonionemus* predation (number of medusae consumed after 24 h).

### Impact of jellyfish consumption on spider crabs

Thirty—six *Libinia* were obtained to assess the effects of increased *Gonionemus* density on crab predation and mortality. Mean carapace width was 73 mm $\pm$ 9 S.D. Crab size differed between treatments (ANOVA, $P = 0.039$, $F = 3.36$, $df = 3$) and crabs in the 20 *Gonionemus* treatment were significantly smaller than in the 15 *Gonionemus* treatment (Tukey's HSD test, $P < 0.05$), but none of the other pairwise comparisons of *Gonionemus* density treatments differed significantly. As in the 2016 trials, jellyfish consumption began in the first few minutes, and was at or near 100% after 24 h for many crabs in all *Gonionemus* density treatments (Fig. 4; suppl. video https://figshare.com/articles/Predation_on_the_clinging_jellyfish_Gonionemus_sp_by_the_spider_crab_Libinia_dubia/5306101, Data S1). We also found that *Libinia* mortality increased as *Gonionemus* density increased, and 100% of the crabs died at the highest *Gonionemus* density treatment (Fig. 6). None of the crabs in the crab-only controls died, and none of the *Gonionemus* in the jellyfish-only controls died.

*Gonionemus* abundance was estimated on July 19, 2017, August 7, 2017, and August 11, 2017 and was 310 (after 60 min of netting), 39 (after 45 min of netting), and 19 (after 45 min of netting) medusae per 3 m $\times$ 3 m search area, respectively. These values do not represent absolute numbers of *Gonionemus* in the search areas and are based on different amounts of search efforts. Rather these values should be considered catch per unit effort estimates and represent a minimum quantity (i.e., there were likely more *Gonionemus* in the search areas, but not less). As *Gonionemus* is primarily sedentary we do not expect that there was influx into the search area from outside the search area over our search periods.

### DISCUSSION

We documented a novel trophic interaction between native crabs and a cryptogenic hydrozoan jellyfish, that may indirectly facilitate dominance of a highly destructive invasive crab in ecologically sensitive eelgrass meadows. Our results are the first example that we

are aware of that demonstrates predation on hydrozoan medusae by crabs. Also, the toxic effects of the jellyfish on the native crabs, coupled with lack of jellyfish consumption (and accompanying toxic effects) by the invasive crabs, provides a mechanism for an indirect, but potentially significant ecological impact on eelgrass communities. The native *Libinia* and invasive *Carcinus* co-exist in eelgrass meadows; however, *Carcinus* can be very destructive to eelgrass shoots (*Garbary et al., 2014*; *Neckles, 2015*; *Matheson et al., 2016*). Both *Libinia* and *Carcinus* have similar diets—both are generalists that prey on a wide variety of organisms (*Aldrich, 1974*; *Grosholz & Ruiz, 1996*). *Gonionemus* thus has the potential to promote *Carcinus* populations by inducing mortality in a native competitor.

While our study was based on laboratory observations, it is very likely that *Libinia* is preying on *Gonionemus* in the field. *Libinia* and *Gonionemus* occupy the same eelgrass microhabitat. In contrast to most jellyfish which are found in the water column, *Gonionemus* medusae spend most of their time attached to eelgrass, in particular near the bottom of the eelgrass where they would be most susceptible to crab predation. Even if the medusae were to cling to the middle or upper part of the eelgrass blades, *Libinia* has the ability to climb (D Grunden & M Carman, pers. obs., 2017). While our field *Gonionemus* density counts do not reflect absolute densities, they do document a minimum baseline that establishes that our laboratory treatments were realistic. It is very likely that *Libinia* encounters far more than 30 *Gonionemus* individuals (as in our maximum *Gonionemus* density treatment, which resulted in 100% mortality) in a 24-hour period, especially at the height of the *Gonionemus* season in July.

It is possible that in the field, given a variety of prey options, that *Libinia* would be less likely to consume large numbers of *Gonionemus* that would have toxic effects. However, our observations showed the crabs had no reluctance in consuming the jellyfish once they were encountered (link to supplemental video https://figshare.com/articles/Predation_on_the_clinging_jellyfish_Gonionemus_sp_by_the_spider_crab_Libinia_dubia/5306101), and consumption of large numbers of jellyfish may not be necessary to elicit a fatal or even a debilitating sublethal effect, as seen by the elevated mortality rate in our lower density trials.

Our results suggest conflicting observations that crab size might be a factor in *Gonionemus*—related crab mortality. In our 2016 trials where 5 *Gonionemus* were offered to each crab, crab mortality was inversely related to crab size category. We did not evaluate possible trial date effects, but note that crabs collected at all 2016 trial dates were represented in each size category. In our 2017 *Gonionemus* density trials, we found that crabs in the 20 *Gonionemus* density treatment were significantly smaller than in the 15 *Gonionemus* density treatment, but this group suffered twice the mortality rate (66.7%) than the 15 *Gonionemus* density treatment (33.3%). However, any potentially beneficial size effects were likely over-ridden by the increase in jellyfish consumption. Thus, the possible relationship between crab size and *Gonionemus*—induced mortality needs further evaluation.

Toxicity may vary between jellyfish individuals and individual crab reactions to the jellyfish toxins may also vary (as they do in humans; *Otsuru et al., 1974*; *Yakovlev & Vaskovsky, 1993*). Given that in some human cases, a sting caused by a single medusa is sufficient to cause extreme pain (*Otsuru et al., 1974*; D Grunden pers. obs., 2008; M Carman pers. obs., 2013) it seems possible that similarly, consumption of even a single medusa by a

crab could have a significant negative effect. In humans, symptoms, which are non-lethal, can persist for a few days (*Yakovlev & Vaskovsky, 1993*); however, human studies may not be directly applicable to crabs. Determining the type, duration, and impact of sublethal effects of *Gonionemus* consumption on crabs would be an interesting future direction. Actual predation rates on *Gonionemus* in the field are hard to assess as the jellyfish lack resistant parts that could be identified in crab gut content analyses (*Arai, 2005*). Molecular probes, however, have great potential to identify prey items in guts that are not otherwise observable (e.g., *McInnes et al., 2017*), and should be considered in future work.

Cnidarian jellyfish predators include sea turtles, fish, molluscs, chaetognaths, ctenophores, and other cnidarians (*Arai, 2005*; *Ates, 2017*). Most of these examples involve predation on scyphozoan jellyfish, but predators of hydrozoan jellyfish (inclusive of siphonophores and *Velella* hydroids) include fish (e.g., *Brodeur, Lorz & Pearcy, 1987*); birds (*McInnes et al., 2017*); hyperiid amphipods (e.g., *Sheader & Evans, 1975*; *Williams & Robins, 1981*); shrimp (*Heffernan & Hopkins, 1981*; *Roe, 1984*; *Nishida, Pearcy & Nemoto, 1988*; *Moore, Rainbow & Larson, 1993*); barnacles (*Bieri, 1966*); spiny lobster phyllosoma larvae (*Wakabayashi et al., 2012*); nudibranchs and heteropods (*Sentz-Braconnot & Carre, 1966*; *Seapy, 1980*); scyphozoan jellyfish (*Purcell, 1991a*; *Purcell, 1997*; *Båmstedt, Ishii & Martlnussen, 1997*; *Arai & Jacobs, 1980*); and even other hydrozoans (*Arai & Jacobs, 1980*; *Purcell, 1981*; *Purcell, 1991b*). The only example of crab predation on a hydrozoan that we could find, however, is the Dungeness crab *Cancer magister* Dana 1852; who, as planktonic larvae, feed on the planktonic hydroids of *Velella* (*Wickham, 1979*).

A small number of jellyfish—crab interactions have been reported for scyphozoan jellyfish (reviewed in *Moyano et al., 2012*; *Ates, 2017*) and ctenophores (*Esser, Greve & Boersma, 2004*). Most of these relationships are symbiotic, where the crabs are associated with scyphomedusae and may benefit from dispersal. Intriguingly, many of the crabs involved in these associations belong to the genus *Libinia*. A small subset of these crab-jellyfish associations involves predation or partial predation on the jellyfish, as opposed to a symbiotic relationship. These include: *Libinia dubia* feeding on the sea nettle *Chrysoara quinqecirrha* Desor 1848 (*Phillips, Burke & Keener, 1969*), the cannonball jellyfish *Stomolophus meleagris* Agassiz, 1862 (*Shanks & Graham, 1988*; *Tunberg & Reed, 2004*), and the moon jellyfish *Aurelia aurita* Linnaeus 1758 (*Jachowski, 1963*); and the graceful crab *Cancer gracilis* Dana 1852 feeding on the moon jellyfish *Aurelia labiata* Chamisso & Eysenhardt 1821 (*Towanda & Thuesen, 2006*). Also, *Carcinus maenus* consumes at least some gelatinous zooplankton in its native European range. *Esser, Greve & Boersma (2004)* describe *C. maenus* predation on the ctenophore *Pleurobrachia pileus* Müller 1776 in the North Sea, particularly when the ctenophores approach the seafloor, and *Lauckner (1980)* reported observations of *Carcinus maenus* consuming tissue of the moon jelly *Aurelia aurita* in the Baltic Sea. *Sweetman et al. (2014)* reported that deep sea galatheid crabs could consume dead scyphomedusan carcasses that originated in shallower water (i.e., "jelly falls").

In addition to being unusual, the relationship between *Libinia* and *Gonionemus* may be shaped by the presence of especially toxic *Gonionemus* lineages (*Govindarajan & Carman, 2016*; *Govindarajan et al., 2017*). We observed *Gonionemus*-induced mortality in *Libinia*

at *Gonionemus* numbers lower than what we expect the crabs encounter in the field. The hard shells of the crabs probably provided protection from *Gonionemus* stings upon initial contact with the jellyfish. However, the soft interior tissues are more likely to be vulnerable. It is interesting to note that inadvertent human consumption of jellyfish on edible seaweed likely also results in toxic effects similar to external stings (*Otsuru et al., 1974*); although again, the mechanisms underlying the toxic effects may differ between crabs and humans.

The readiness of *Libinia* to unhesitatingly consume jellyfish which may result in their death is consistent with the hypothesis of a recent introduction of a highly toxic strain (*Govindarajan & Carman, 2016*). It seems likely that consumption of toxic jellyfish would exert a strong selection pressure on the consumers, that over time would result in the evolution of jellyfish avoidance or toxin tolerance mechanisms, or the disappearance of crabs from jellyfish habitats. Toxin tolerance mechanisms are possibly present in other crustacean predators of jellyfish. *Wakabayashi et al. (2012)* observed that spiny lobster phyllosoma larvae consumed both highly venomous jellyfish species (the Portuguese man-of-war *Physalia physalis* Linnaeus, 1758, the box jellyfish *Carybdea rastonii* Haacke, 1886, and the Japanese sea nettle *Chrysaora pacifica* Goette, 1886) as well as less toxic species, with no ill effects described in any cases. They speculate that tolerance to jellyfish toxins may have evolved for open ocean predators, where non-toxic prey items are relatively more scarce. However, their trials involved consumption of single jellyfish, and it is possible that increasing consumption could lead to mortality. In any case, *Libinia*'s eager consumption of *Gonionemus* coupled with its lack of tolerance to its toxin suggests that this interaction is recent.

Records of *Gonionemus* sightings and stings also support the hypothesis that the *Libinia*—toxic *Gonionemus* interaction may be new. Our study site, Farm Pond, is located close to Sengekontacket Pond, where a less toxic *Gonionemus* population that was regularly accessed by jellyfish collectors was known to exist for decades before the first stings were recorded (*Govindarajan & Carman, 2016*). However, debilitating stings have occurred only in the past few years in Farm Pond (*Govindarajan & Carman, 2016*; and directly to D Grunden & M Carman), suggesting the arrival of a new, highly toxic form. While we did not quantify the toxicity of the jellyfish used in our experiments, *Govindarajan et al. (2017)* found that Farm Pond primarily contained a mitochondrial haplotype that is found in other Northwest Atlantic locations where stings have occurred.

Our finding that in contrast to *Libinia*, *Carcinus* does not consume *Gonionemus* has significant implications for eelgrass ecosystem health. *Carcinus* is native to Europe, where a less toxic form of *Gonionemus* (*Gonionemus vertens* A. Agassiz, 1862) is thought to be introduced (*Edwards, 1976*; *Bakker, 1980*). Thus, it may not have historically been exposed to selective pressure by the more toxic form that would explain its avoidance of *Gonionemus* consumption. Future experiments should test whether *Carcinus* consumes European *Gonionemus vertens*, or if predation-induced *Libinia* mortality is determined by the lineage of its *Gonionemus* prey (as the lineages vary in their toxicity to humans; *Naumov, 1960*; *Govindarajan & Carman, 2016*).

The difference we observed between *Carcinus* and *Libinia* might instead be due to a stronger pre-existing preference of *Libinia* to consume jellyfish. While both *Carcinus* and

*Libinia* have broad and overlapping diets, preferences differ between the two species. *Carcinus* tends to be more aggressively predatory (*Ropes, 1968*; *Griffen, 2014*), although it can shift towards herbivory in response to competition (*Griffen, Guy & Buck, 2008*). Spider crabs, such as *Libinia*, tend to be omnivorous scavengers, often feeding on carrion and algae (*Wicksten, 1980*; *Stachowicz & Hay, 1999*). As noted earlier, *Libinia* is known to consume scyphozoan jellyfish (that presumably lack the extreme toxic effects of *Gonionemus*) (Philips, Burke & Keener, 1969). We also observed *Gonionemus* predation by one out of the 2 blue crabs that we evaluated. While blue crabs were too rare to evaluate further, it is interesting that like *Libinia*, they have been reported to consume scyphozoan jellyfish (*Farr, 1978*).

Our results also have implications for a broader understanding of invasive species impacts. In addition to having direct effects on native species, for example through competition or predation, invasive species can have indirect effects, but these are less explored (*White, Wilson & Clarke, 2006*). Indirect effects occur when one species affects another via a third species (*Wootton, 1994*), and include apparent competition, indirect mutualism/commensalism, trophic cascades, and exploitative competition (*White, Wilson & Clarke, 2006*). We have identified a unique indirect mechanism by which a cryptogenic jellyfish can potentially increase the abundance of an aggressive and highly destructive invasive species, *Carcinus*. Both *Carcinus* and *Libinia*, overlap in habitat and as generalists, they are both known to feed on a broad array of other species, and so they are likely competing for common prey resources. Thus *Gonionemus*-induced mortality of *Libinia* could benefit *Carcinus* populations by increasing prey abundance. Given the highly negative impact of *Carcinus* to sensitive eelgrass systems, it is important to evaluate this hypothesis as well as identify other ecosystem effects of *Gonionemus* (e.g., its role as a predator, as well as prey).

## ACKNOWLEDGEMENTS

We thank Dann Blackwood (USGS Woods Hole) for assistance with photography and video, Jason Mallory (Oak Bluffs Shellfish Department) and Kallen Sullivan (Oak Bluffs Shellfish Department) for assistance in conducting the experiments, Pam Polloni (WHOI) for providing helpful comments on the manuscript, Dale Calder (Royal Ontario Museum) for taxonomic advice, and the Martha's Vineyard Shellfish Group, Inc. (leased from the Massachusetts Division of Marine Fisheries) and the Oak Bluffs Shellfish Department for providing laboratory space.

### Funding

This work was supported by the Oak Bluffs Community Preservation Committee under Grant 45908900; Oak Bluffs Community Preservation Committee under Grant 45785700; USGS-WHOI Cooperative Program under Grant 48010601, the Adelaide M. and Charles B. Link Foundation, and the Kathleen M. and Peter E. Naktenis Family Foundation. The

funders had no role in study design, data collection and analysis, decision to publish, or preparation of the manuscript.

### Grant Disclosures
The following grant information was disclosed by the authors:
Oak Bluffs Community Preservation Committee: 45908900, 45785700.
USGS-WHOI Cooperative Program: 48010601.
The Adelaide M. and Charles B. Link Foundation.
The Kathleen M. and Peter E. Naktenis Family Foundation.

### Competing Interests
The authors declare there are no competing interests.

### Author Contributions
- Mary R. Carman conceived and designed the experiments, performed the experiments, analyzed the data, contributed reagents/materials/analysis tools, reviewed drafts of the paper.
- David W. Grunden performed the experiments, contributed reagents/materials/analysis tools, reviewed drafts of the paper.
- Annette F. Govindarajan conceived and designed the experiments, analyzed the data, wrote the paper, prepared figures and/or tables, reviewed drafts of the paper.

### Field Study Permissions
The following information was supplied relating to field study approvals (i.e., approving body and any reference numbers):

Co-author Dave Grunden as a Shellfish Constable in Massachusetts has the authority from the state of Massachusetts to collect the animals used in this study (Chapter 130 Section 98).

### Data Availability
Carman, Mary; Grunden, Dave; Govindarajan, Annette (2017): Predation on the clinging jellyfish *Gonionemus* sp. by the spider crab *Libinia dubia*. figshare. https://doi.org/10.6084/m9.figshare.5306101.v1.

### Supplemental Information
Supplemental information for this article can be found online at http://dx.doi.org/10.7717/peerj.3966#supplemental-information.

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
