# Peer review of "Species–specific crab predation on the hydrozoan clinging jellyfish Gonionemus sp. (Cnidaria, Hydrozoa), subsequent crab mortality, and possible ecological consequences"

_PeerJ, doi:10.7717/peerj.3966_

## Round 0.1 · original submission · Minor Revisions

I have heard back from two reviewers, both of whom felt your paper is a worthy contribution to PeerJ. There are a number of minor comments to be addressed, but no major issues that would require extensive revision, and hence my decision is "minor revisions" are needed.

·

Basic reporting

This MS by Carman, Grunden and Govindarajan is excellent, providing new data and observations on predation on the hydrozoan jellyfish Gonionemus by crab species both living in eelgrass habitats in Massachusetts, USA.

The MS is written clearly, with experiments, results and implications expressed in sufficient detail and fairly. Main conclusions include that the native majoid crabs Libinia dubia readily consume Gonionemus and that mortality of L. dubia increases in experiments when more medusae are consumed. In contrast, the introduced crab Carcinus maenus avoids consuming the jellyfish. Anecdotal evidence is presented that Callinectes sapidus may consume Gonionemus, but C. sapidus was not common enough to assess this relationship in any detail.

All told, these results are relevant because L. dubia and C. maenus live in the same habitat and consume many of the same prey items, suggesting that the differential feeding bahaviors on Gonionemus benefits the non-native C. maenus.

The complicating detail in this paper is that recent work has shown that the Gonionemus jellyfish likely in this region may include lineages that are particularly noxious to humans and that have recently been introduced. Some care should be taken to make clear that toxicity to humans is not necessarily a good proxy for toxicity to crabs, or evidence to the contrary should be presented.

Also, it may be helpful to explicitly lay out potential future work investigating whether there is differential impact when Libinia feed on the different lineages of Gonionemus.

Experimental design

The experiments are fairly straight forward.

Validity of the findings

The experiments are fairly straight forward, consistently showing that: 1) more medusae are consumed by L. dubia over time and in higher densities, and; 2) mortality increases as more medusae are consumed. The apparent inverse relationship between crab size and mortality (with smaller crabs being less likely to suffer lethal effects) is difficult to understand and the authors correctly assert that more evidence is needed to understand whether this relationship is real or not.

Additional comments

A few notes/suggestions, all minor.
1) I do not necessarily agree that "jellyfish" refers to gelatinous zooplankton. Perhaps this is true of the word jelly or jellies, but I tend to think of jellyfish referring to only the medusa phase of medusozoan species that have a medusa phase. In any event, it is a common name and it is unclear how that introductory sentence helps the overall narrative of this paper. It is not a big deal, but the authors could use the term "gelata" , as introduced by Haddock (Haddock, 2004 in Hydrobiologia) in order to move the focus to the medusa-bearing groups.
2) Line 223: "Even if the medusae were cling. . ." should have "were" deleted.
3) Lines 246-250. It might be worth providing a bit of caution into this discussion. Crabs and humans are quite different organisms and there is no evidence that the toxins behave similarly on the two groups. I think a caveat to this effect might be warranted. It is interesting to note the individual variation in human response to Gonionemus venom, and the fact that a single sting can elicit symptoms in a human, but this is merely suggestive of possibilities in regards to the crabs.
4) The discussion of jellyfish predators starts (lines 257-258) with a partial list (sea turtles, fish, molluscs, chaetognaths, ctenophores, and other cnidarians) and that through me off a bit. But it goes on to become more complete. Phyllosoma larvae should be added to the list (Biol Bull. 2012. 222(1):1-5. Predation by the phyllosoma larva of Ibacus novemdentatus on various kinds of venomous jellyfish.) and I have also seen images of galatheid crabs feeding on medusae in deep reef settings. I was unable to find anything in a quick literature search, but the authors might consider checking that.
5) Lines 288-290 states "The hard shells of the crabs probably provided protection from Gonionemus stings upon initial contact with the jellyfish. However, the soft interior tissues appear to be vulnerable." Is there really any evidence that the stings do not occur through the handling and manipulation of the jellies, rather than the consumption? I suspect that the authors have this correct, but it might be more fair to state " However, the soft interior tissues are more likely to be vulnerable."
6) Lines 308-312. Here again the authors suggest that what we know about Gonionemus toxicity to humans may be directly relevant to toxicity in crabs (when discussing Carcinus avoidance of Gonionemus). It could also be the case that Gonionemus in Europe is toxic to Carcinus even if the Gonionemus in Europe isn't particularly noxious to humans.
7) The authors may want to lay how to test (and the importance of testing) whether the less toxic lineage of Gonionemus (to humans) is also less toxic to Libinia.

·

Basic reporting

This is a straightforward experimental paper on the effect of Gonionemus cf. murbachii on Libinia dubia and other crab species.

Experimental design

Very clear.

Validity of the findings

The results are sound and worth publishing.

---

## Round 0.2 · accepted · Accept

Your paper has been well revised and is now ready for publication. I look forward to seeing the published version.

·

Basic reporting

All clear, and reviewer comments addressed satisfactorily.

Experimental design

Well done.

Validity of the findings

Conclusions are well founded.

Additional comments

All reviewer comments have been addressed and the MS is ready to be published.